# Gastrointestinal Involvement in Children with Systemic Lupus Erythematosus

**DOI:** 10.3390/children10020309

**Published:** 2023-02-06

**Authors:** Angela Mauro, Teresa Giani, Clelia Di Mari, Martina Sandini, Antonella Talenti, Valentina Ansuini, Luigi Biondi, Giovanni Di Nardo, Luca Bernardo

**Affiliations:** 1Pediatric Rheumatology Unit, Department of Childhood and Developmental Medicine, Fatebenefratelli-Sacco Hospital, Piazzale Principessa Clotilde, 20121 Milano, Italy; 2Department of Pediatrics, Meyer Children’s Hospital, 50139 Firenze, Italy; 3Faculty of Medicine and Psycology, Sapienza University of Rome—NESMOS Department, Sant’Andrea University Hospital, Via Grottarossa 1035-1039, 00189 Roma, Italy

**Keywords:** childhood-onset systemic lupus erythematosus, children, gastrointestinal involvement, autoimmune hepatitis, pancreatitis, mesenteric vasculitis, peritonitis, intestinal pseudo-obstruction, review

## Abstract

Systemic lupus erythematosus (SLE) is a systemic autoimmune disorder. When it presents before the age of 18 years (childhood-onset systemic lupus erythematosus, cSLE), the disease course tends to be more severe with a higher rate of organ involvement and requires an early diagnosis. Gastrointestinal involvement in cSLE is rare and scarcely reported in the literature. Any organ of the gastrointestinal system may be affected, either as a direct consequence of the disease, as a subsequent complication, or as an adverse drug event. Abdominal pain is the most common GI symptom, it can be diffuse or well localized, and can underline different conditions such as hepatitis, pancreatitis, appendicitis, peritonitis, or enteritis. cSLE may have an alteration of the intestinal barrier with features of protein-losing enteropathy or, in genetically predisposed patients, may develop associated autoimmune disorders such as Coeliac Disease or Autoimmune Hepatitis. The aim of this manuscript is to provide a narrative review of gastrointestinal manifestations in cSLE focused on hepatic, pancreatic, and intestinal involvement. A comprehensive literature search based on the PubMed database was performed.

## 1. Introduction

Systemic lupus erythematosus (SLE) is a systemic autoimmune disease that can involve different organs and systems. The prevalence of SLE changes in relation to gender and ethnicity, with the highest expression in African American women. Up to a quarter of the cases develop during childhood, usually after the first decade of life. When the disease onset is before 18 years of age, SLE is defined as childhood-onset SLE (cSLE). In general, cSLE has a more severe disease course, with a higher rate of organ involvement when compared to adult onset. An early diagnosis is crucial to improve morbidity and mortality [1].

Classification criteria were firstly published in 1982, and then revised in 1997 by the American College of Rheumatology (ACR). In order to classify a patient as affected by SLE, 4 out of 11 defined criteria should be satisfied [2,3]. In 2012 a new proposal for SLE classification criteria was published by the Systemic Lupus International Collaborating Clinics (SLICC). The fulfillment of at least 4 out of 17 criteria is required, with at least 1 clinical and 1 laboratorial criterion or a renal histology that is compatible with lupus nephritis plus ANA or anti-DNA positive antibodies [4].

In pediatric age, the SLICC criteria seem to have a higher sensitivity and accuracy with respect to the ACR ones, with a comparable specificity (see Table 1) [5].

Constitutional symptoms, such as fever and fatigue, are often observed at the disease onset and the clinical picture is usually characterized by cutaneous and musculoskeletal symptoms. However, the prognosis mainly depends on renal failure, neurologic disease, and pulmonary hemorrhage.

Gastrointestinal (GI) involvement is rare in cSLE and scarce data have been reported. Any part of the GI system may be involved, and symptoms can be the direct consequence of the primary disease, coincidentally associated with the primary disease, a subsequent complication of the primary condition, or an adverse treatment effect [6]. GI involvement can occur both at the disease onset and during the disease course [7,8].

The most common GI symptom is abdominal pain, which can be either diffuse or well localized and can be associated with different conditions such as pancreatitis, appendicitis, peritonitis, cholecystitis, or enteritis [9,10].

Sönmez et al. retrospectively identified 19/69 cSLE patients with a GI involvement, of which 3/19 at the disease onset: in 9/19 cases it was a direct SLE consequence expressed as hepatitis or enteritis, in 4/19 cases it was associated with SLE as hepatomegaly, hypertransaminasemia, and steatosis, in 5/19 patients it appeared as a drug adverse event. One child developed appendicitis, which was not clearly related to cSLE. Abdominal pain, diarrhea, hypertransaminasemia, and hepatomegaly were the most common GI manifestations [11].

Among non-SLE causes, an underlying infection should be ruled out [12,13].

In the present review, we aim to focus on GI involvement in cSLE, offering an overview on the main clinical manifestations, differential diagnoses, and management.

## 2. Methods

This narrative review was performed in accordance with the IMRAD (introduction, methods, results, and discussion) approach [14] and presents a non-systematic analysis of the literature on the GI involvement in children with SLE.

The authors considered original scientific papers, case reports/series, and reviews of major relevance, written in English language, published in the past ten years (2012–2022). PubMed, Scopus, EMBASE, and Web of Science were utilized as electronic databases for this research.

Keywords (alone and/or combined) were used to select relevant articles, in particular: (“Systemic Lupus Erythematosus” AND (“Intestinal Involvement” OR “Intestinal symptoms” OR “Gastrointestinal” OR “Hepatic involvement” OR “Pancreatic involvement” OR “mesenteric vasculitis” OR “peritonitis” OR “intestinal pseudo-obstruction” OR “protein-losing enteropathy”) AND (“Child” OR “Children” OR “Childhood”), (see Figure 1).

The contributions were independently collected by MS, AT, LB and VA, the manuscript was drafted by AM, TG and CDM, reviewed and discussed by AM, GDN and LB.

The final version was then approved by all.

## 3. Hepatic Involvement

Hepatic involvement is rare in children, and it is associated with a higher mortality rate [15].

Hepatitis in cSLE has usually been described as a primary (lupus hepatitis) side effect of the pharmacological treatment, and a consequence of infective complications, related to fatty liver, or as primary autoimmune hepatitis (AIH) [16,17]. Differential diagnosis between primary and secondary hepatitis is complex and it is crucial to start a correct treatment of the underlying disease [18].

Hepatic involvement is often mild and subacute, and it is characterized by an increase in serum transaminase levels. The diagnosis of SLE-related hepatitis is of exclusion.

Tahernia et al. evaluated the frequency and type of hepatic involvement in 138 cSLE patients that were admitted to Children’s Medical Center in Iran, from 2005 to 2014. In 48.5% of the patients, an increase in hepatic enzymes was detected as 8.7%, 5%, and 34.7% abnormal aspartate aminotransferase (AST), alanine transaminase (ALT), and both, respectively, and categorized on the base of their levels in less than 100 U/mL (23.1%), between 100 and 1000 U/mL (23.1%), and more than 1000 U/mL (2.1%). Although about 50% of cSLE patients had hepatic involvement, no significant correlation emerged between the increase of liver enzymes and clear organ damage [19].

### 3.1. Autoimmune Hepatitis in cSLE

Rarely, lupus hepatitis and autoimmune hepatitis coexist (AIH-SLE) in the same patient and the differential diagnosis is often challenging [20].

The occurrence of AIH-cSLE is rare, and its prevalence is still unknown [21,22].

AIH is characterized by chronic liver inflammation. The diagnosis is based on characteristic clinical and laboratory findings as hypergammaglobulinemia, one or more typical autoantibodies, and interface hepatitis on histological examination [23,24]. Compared to lupus hepatitis, AIH–SLE overlap syndrome is more aggressive in terms of histological pattern and prognosis.

Differently from lupus hepatitis, HIA-SLE is characterized by high levels of AST and ALT. In fact, in the study that was performed by Takahashi, who studied liver dysfunction in 123 adult patients with SLE in terms of clinical features, laboratory, differential diagnosis, and prognosis, where liver dysfunction caused by AIH was associated with higher levels of AST, ALT, and alkaline phosphatase (ALP) than lupus hepatitis with a worse long-term prognosis [18]. Moreover, the literature described cases in which AIH sometimes progressed to fulminant hepatic failure [25].

Typical histological features of AIH are interface hepatitis, characterized by an inflammatory infiltration of lymphocytes and plasma cells, which invade adjacent parenchyma; panlobular hepatitis with bridging necrosis; and, less common, an emperipolesis and liver cell rosette formation [26]. Lupus hepatitis is described as lobular hepatitis, with atrophy and necrosis of the central hepatic cells, fatty infiltration, and inflammatory lymphocytes infiltrates [27,28].

A scoring system has been proposed by the International Autoimmune Hepatitis Group (IAIHG) and the Systemic Lupus International Collaborating Clinics (SLICC) to differentiate AIH and lupus hepatitis (see Table 2) [29,30,31].

There are also differences in terms of treatment. AIH-SLE overlap syndrome patients need an aggressive therapy with high-dose of steroids (prednisone 1–2 mg/kg daily) for up to 2 weeks, while lupus hepatitis is treated with non-steroidal anti-inflammatory drugs, corticosteroids, and immunomodulators [32].

In a multicenter study, Balbi et al. described demographic data, clinical manifestations, treatments, and outcomes in a large cSLE population with AIH [33]. They retrospectively analyzed 847 patients with cSLE. A total of 7/847 presented AIH, confirmed by biopsy, and they were all adolescents. AIH occurred before or at cSLE diagnosis and it was associated with mild liver manifestations. Hepatomegaly was the most typical manifestation and none of these patients had severe hepatic manifestations such as hepatic failure, portal hypertension, or cirrhosis. No differences emerged in terms of treatments and systemic involvement such as splenomegaly, serositis, musculoskeletal, neuropsychiatric, and renal manifestations with respect to patients without hepatic involvement. Antinuclear antibodies were detected in all AIH patients, 43% also had anti-smooth muscle antibodies, and all patients were seronegative for anti-liver kidney microsomal antibodies.

Different drugs that are usually prescribed in SLE such as methotrexate, hydroxychloroquine, and azathioprine can be responsible for liver damage [34]. Even if not directly comparable to the pediatric population, data that are related to a large cohort of 1533 adult SLE patients reported by Huang et al., suggested that metabolic abnormalities such as obesity, hypertension, and drug-related toxicity may provide hepatotoxicity more than SLE-related factors [35].

### 3.2. Non-Alcoholic Fatty Liver in cSLE

Non-alcoholic fatty liver disease is a complex dysmetabolic process including different conditions ranging from steatosis to end-stage liver disease. This is a growing condition in developed countries, where it affects 3–10% of the general pediatric population, and up to 34% of those that are affected by obesity [36,37]. Sönmez et al. found that among 69 pediatric SLE patients, 19 (27.5%) of them presented a GI involvement and one young patient had steatosis [11].

In a study that was conducted in an adult population, Mackay et al. observed minimal histological changes in 11 SLE patients including fatty liver, portal fibrosis, and mild to moderate portal infiltrate [38]. The widely accepted theory of “multiple-hit model” considers the implication of different factors in the pathogenesis of this condition such as hepatocytes accumulation of triglycerides, and insulin resistance [39]. In SLE, chronic inflammation, drug-related liver injury, exposure to corticosteroids, and changes in the gut microbiome represent the main causes involved [40].

An early identification and a focused intervention on those factors that can be modified, such as weight and lifestyle, are of primary importance. Non-invasive radiological methods are useful in the screening of hepatic steatosis, even if they could be insensitive in the presence of a limited damage. In the early phases of the disease, the levels of ALT and AST also cannot accurately reflect the liver damage, and liver biopsy is the only exam that may definitively confirm the diagnosis [41,42,43].

### 3.3. Autoimmune Cholestatic Liver Disease in cSLE

Primary biliary cholangitis (PBC) is an autoimmune chronic cholestatic liver disease characterized by a progressive destruction of the small biliary ducts. Limited reports have described the association with SLE, that seem to be even rarer in pediatric age [44,45,46,47].

In a large retrospective Chinese study, based on the data of 769 adult PBC patients, 26 (3.4%) were affected by SLE [45].

Even if the prevalence of this overlapping is low, the presence of SLE seems to have a negative impact on the prognosis of PBC.

## 4. Pancreatic Involvement

Acute pancreatic inflammation is associated with exocrine pancreas damage and autodigestion, resulting in intra-acinar enzymes and systemic cytokines release [48]. The pathogenic mechanism of acute pancreatitis (AP) in patients with SLE is unclear, and it is supposed to involve different mechanisms. Autoantibodies, immune complexes deposition, complement activation, vasculitis, intimal thickening, vessel occlusion, and abnormal immune response of pancreatic cells may all be involved [49,50]. The overall incidence of AP in children is estimated to be 3.6–13.2/100,000, while the prevalence of AP in SLE ranges from 0.7 to 4% of the cases, with a higher involvement among young female patients [51,52]. Only limited data have been published regarding the pediatric age. In children, the disease seems to be more severe, with a higher mortality rate than in adults, especially if thrombocytopenia, leukopenia, hypoalbuminemia, or hematuria are present [49,50,51,52,53]. Marques et al. enrolled 852 pediatric SLE patients from 10 pediatric rheumatology centers in order to analyze and classify the pancreatic involvement according to the International Study Group of Pediatric Pancreatitis. Among these children, AP was diagnosed in 22, and in half of the cases it resulted to be aggressive. In 20/22 children, the disease course was monophasic, and in 2/22 it was recurrent [54,55]. In adult SLE patients, the recurrence rate ranges from 11 to 43% of the cases on the basis of different studies [54,55,56].

AP may cause local and/or systemic complications. In adult SLE patients, the most frequently reported complications are the respiratory, cardiovascular and renal insufficiency, ascites, and peritoneal hemorrhage [53,57,58,59]. AP usually occurs in patients with a severe disease and during flares. Some reports suggested a possible association with macrophage activation syndrome (MAS), a potential SLE complication, pointing out the pancreas as a potential target organ of this syndrome [60,61]. In a case-based literature review including 87 pediatric patients with SLE and pancreatitis, 52.86% of them had MAS at the same time [62]. The diagnosis of AP is based on the presence of two out of the three INSPPIRE (International Study Group of Pediatric Pancreatitis: in search for a cure) and Atlanta criteria including one clinical, one biochemical, and one imaging criteria [63]. AP is suspected in the presence of abdominal pain that often is poorly localized and exacerbated by supine positioning. Nausea, vomiting, and epigastric tenderness can also be present together with more typical SLE features such as fever, musculoskeletal and cutaneous symptoms, or manifestations that are linked to major organ system involvement [64]. Clinical presentation may range from a subclinical pancreatitis, characterized by high levels of pancreatic enzymes with no clear symptoms, to a severe pancreatitis, with organ failure and pancreatic necrosis.

Laboratory exams may reveal elevated serum amylase or lipase; when their increase is at least three times the upper normal value, it is considered consistent with AP. Although both enzymes increase early in the disease course, lipase is more specific as it is primarily released from the pancreas. Lipase increases during the first 6 h and then persists for the following days. Amylase has both a pancreatic and a salivary secretion and usually shows a more dynamic course, picking faster and normalizing within the first 24 h [65].

Imaging can confirm AP diagnosis in the presence of unspecific clinical or biochemical signs; it also allows the exclusion of other causes and may detect the parenchymal necrosis [66]. Ultrasounds represent the first and basic imaging test, allowing the detection of increased volume and a reduction of echogenicity, although interfering structures may limit the anatomical assessment; in these cases, contrast enhanced computed tomography is helpful [67].

Although rare, AP often develops during the early phases of SLE, sometimes as a presenting manifestation [68].

Defects, trauma, or other mechanical causes need to be excluded, as well as infections, toxic and metabolic etiologies such as gallstones, hyperlipidemia, and those that are related to chronic renal disease [69].

Together with supportive care, corticosteroids are usually part of the first strategy and, in severe cases, cyclophosphamide or plasmapheresis may represent efficacious alternatives [56].

Drug-induced pancreatitis, although rare, should be taken into account. Some drugs that are used to treat SLE, such as corticosteroids, azathioprine, or valproic acid may cause pancreatitis as an adverse reaction. A careful history of medications including their duration of use is recommended. In case of iatrogenic pancreatitis suspicion, a prompt interruption of the presumed drug should be considered [70].

## 5. Intestinal Involvement

### 5.1. Lupus Mesenteric Vasculitis

Lupus enteritis or lupus mesenteric vasculitis (LMV) represents the most common kind of GI involvement in cSLE [71]. It consists of an inflammatory condition that is caused by immune complex deposition with consequent thrombosis and necrosis of the intestinal vessels [72,73,74].

LMV can present with different signs and symptoms ranging from mild abdominal pain to nausea, vomiting, diarrhea, and up to severe acute abdominal pain associated with gastrointestinal bleeding, intestinal necrosis, and perforation. Jejunum and ileum are the most commonly affected as LMV often involves the superior mesenteric artery [75,76]. The oral cavity can also be involved, mostly with painless ulcerations that are usually located in the hard palate; hyperkeratotic lesions and lichenoid infiltrates in the shape of oral macules and papules have also been described [77].

Unfortunately, no association with specific autoimmune profiles exists but occasionally patients with LMV present antiphospholipid antibodies and an increase of D-Dimer values [78].

Abdominal computed tomography scan (abdominal-CT scan) represents the examination of choice in LMV detection. It usually reveals a diffuse or focal bowel wall thickening of at least 3 mm with peripheral rim enhancement (target sign), engorgement of the mesenteric vessels (comb sign), and increased attenuation of mesenteric fat and intestinal dilatation [79]. Clinical and imaging features contribute to LMV diagnosis based on the following two criteria: (1) symptoms are controlled by steroids or immunosuppressive therapy, and evidence of bowel dilatation or diffuse or focal bowel wall thickening; and (2) abdominal-CT scan with contrast showing at least three or more of the following signs: thickening of the intestinal wall, segmental intestinal dilatation, mesenteric vascular enhancement, and blurred mesenteric fat [80].

LMV is usually associated with an active involvement of other organs, although it is rarely an isolated manifestation; in 65% of cases it is concomitant with lupus nephritis. As a consequence of the combined involvement, patients with LMV often present increased disease activity scores, for example on the SLE disease activity index (SLEDAI) or on the British Isles Lupus Assessment Group’s (BILAG) disease activity index [72,76,77,81,82].

Liu et al. described three cases of cSLE that were associated with LMV, admitted to the hospital for abdominal pain and vomiting whose diagnosis was confirmed by abdominal-CT scan. They all presented an active, moderate to severe disease and a concomitant involvement of other organs such as ureteropelvic dilatation, hydronephrosis, and proteinuria; one of them also manifested neurological features. Steroids and immunosuppressive treatments were started, with subsequent rapid improvement [83].

LMV usually responds to corticosteroids, which should be started as quickly as possible to avoid complications, such as intestinal wall necrosis and perforation. Immunosuppressive therapy with cyclophosphamide or mycophenolate mofetil should be associated in case of a severe or resistant course. Complicated cases may require a surgical intervention [72,84,85].

Chowichian et al. described a case of a 12-year-old girl with recurrent abdominal pain associated with nausea, vomiting, and diarrhea as presenting symptoms of her cSLE. An abdominal-CT scan showed small bowel thickening and fat stranding in the left lower abdomen while a colonoscopy identified an intestinal vasculitis. A treatment with prednisone and hydroxychloroquine was started with a subsequent progressive resolution of GI manifestations [86].

### 5.2. Lupus Peritonitis and Intestinal Pseudo-Obstruction

Other intestinal manifestations are very rare but well described in cSLE, such as lupus peritonitis (LP) and intestinal pseudo-obstruction (IPO).

LP can present during disease flares and it clinically manifests with abdominal pain and exudative, non-infected ascites. Sometimes LP is associated with other serositis such as pleuritis or pericarditis [87].

IPO is characterized by signs and symptoms of intestinal obstruction in the absence of a mechanical occlusion. The etiology is unclear, but it seems to be related to a dysfunction of the visceral smooth muscle as a consequence of the enteric nerves or autonomic visceral nervous system damage. Sometimes IPO is associated with uretero-hydronephrosis and, rarely, to esophageal or biliary dilatation [88,89].

IPO can manifest with mild to severe abdominal pain associated with nausea, vomiting, weight loss, and constipation. In contrast to LMV, IPO can also occur during mild SLE activity [90]. At abdominal-CT scan, IPO presents dilated fluid-filled bowel loops, with a thickened bowel wall and multiple fluid levels in the small and/or large bowel. An early diagnosis of this condition and an early treatment with high doses of corticosteroids is important in order to avoid severe complications such as bowel necrosis, bowel perforation, and acute renal failure [91].

### 5.3. Protein-Losing Enteropathy

Lupus protein-losing enteropathy (LUPLE) represents the third most common type of GI involvement in cSLE, following lupus mesenteric vasculitis and lupus intestinal pseudo-obstruction [71,92].

It is more frequent in females and Asiatic children and it can be the presenting manifestation or can occur during the SLE course, associated or not with other GI symptoms [93].

LUPLE consists of an exaggerated protein loss from the intestinal tube due to an impaired intestinal permeability, with consequent hypoalbuminemia and edema, ascites, and diarrhea [94]. In SLE, intestinal vessel permeability is increased in relation to vasculitis and, indirectly, for a vasodilation mediated by complement activation and cytokines [77]. Fecal alpha-1 antitrypsin concentration reflects the intestinal permeability and its increased fecal clearance is used as a biomarker of intestinal protein loss [95]. Technetium 99 m labeled human albumin scintigraphy is able to detect the site of protein leakage, most commonly represented by the small intestine [7]. The treatment consists of corticosteroids and immunosuppressants [65]. Hedrich and colleagues described the case of an 11 month-old female child that was diagnosed with early onset SLE who presented peripheral and periorbital edema. Protein-losing enteropathy was diagnosed because of important non-renal hypoalbuminemia, and increased levels of fecal alpha-1 antitrypsin and calprotectin. Ileocolonoscopy and esophagogastroduodenoscopy showed duodenitis with atrophic or deformed villi and inflammatory infiltrates on histology. Alpha-1 antitrypsin and calprotectin levels normalized under treatment with corticosteroids and azathioprine [96].

### 5.4. Coeliac Disease in cSLE

Coeliac disease (CD) is a systemic disease that is characterized by an anomalous immune-mediated response to gluten intake in genetically susceptible individuals. It causes intestinal enteropathy described as an increased intraepithelial lymphocytes, crypts hyperplasia, and shortened/atrophic intestinal villi (Figure 2) [97].

CD can be associated with a wide range of extra-intestinal signs and symptoms such as iron deficiency anemia, impaired bone metabolism, short stature, elevated liver enzymes, and skin manifestations [98].

The prevalence of CD in children is estimated to range from 0.10% to 3.03% worldwide, with a significantly increasing annual trend [99].

The alteration of the intestinal barrier and a predisposing genetic condition cause CD patients to have a higher risk of developing associated autoimmune disorders such as Type 1 diabetes mellitus, autoimmune thyroiditis, juvenile arthritis, and SLE, compared to healthy controls, with a prevalence ranging from 14% to 27% [100,101,102]. The concomitance of SLE and CD is rare. However, there are some data reporting this association, suggesting that the screening for CD in patients with SLE should be taken into account as clinical features may overlap [103,104,105,106,107,108,109,110]. However, the risk of SLE seems to be increased three-times in CD compared with healthy controls [111]. This is demonstrated by Ludvigsson et al., who investigated the risk of SLE in patients with biopsy-verified CD compared to the general population. In detail, they analyzed 29,048 individuals with biopsy-verified CD (villous atrophy, Marsh 3) from 28 Sweden’s departments with 144,352 healthy controls identified through the Swedish Total Population Register. At most, two individuals with CD out of 1000 could develop SLE in the 10 years following CD diagnosis, but the absolute risk is actually low [112].

Furthermore, a study by Shamseya et al. evaluated the presence of CD serology in a group of 100 SLE adult patients (34.6 ± 9.6 years) who had a c-SLE onset, compared with a control group (n = 40). All the participants were tested for anti-tissue transglutaminase antibodies (tTG) IgA (or tTG IgG, if IgA deficiency was detected). Additionally, anyone who was tTG-positive was tested for anti-endomysium antibodies (EMA) IgA. A total of 10 patients (10%) were both tTG- and EMA IgA-positive compared to controls that were negative. All these 10 cSLE patients underwent upper gastrointestinal endoscopy and the histopathological assessment confirmed CD in six patients, while four cases were defined as latent CD. They concluded that even though cSLE does not seem to be associated with CD during the pediatric age, these patients may be at increased risk later, in their adult life [113].

Moreover, a more recent study that was performed by Soltani et al., including both adults and children, reported a prevalence of biopsy-proven CD in 3% of patients with SLE [114].

Clinically, according to pediatric and adult literature that described patients with SLE—CD overlap, abdominal pain, diarrhea, constipation, loss of appetite, and weight loss were the most common manifestations [115,116].

Crişcov et al. described a case of a 6 year old girl that presented malaise, abdominal pain, loss of appetite, and abdominal distension. After three weeks she developed arthralgias, malar rash, finally being diagnosed with SLE. Based on blood exams (iron deficiency anemia and IgA deficiency) associated with clinical signs, they suspected an overlap of SLE and CD. Diagnosis of CD was confirmed by the presence of an anti IgG tTG antibodies titer of 120 EU/mL and positive small bowel biopsy (IIIB1 stage according to the Marsh classification) [117].

It has been observed that in SLE patients, anti-deamidated gliadin peptides antibodies (DGPs) and immunoglobulin IgA (IgA) seem to be commonly positive [110].

In the study that was performed by Abdelghani et al. involving 24 patients with SLE, DGPs and tTG IgA antibody positivity was detected in seven (29%) and two (8%) patients, respectively, and one (4%) of them was positive for both the DGP and tTG IgA antibody [110].

Dima et al. studied 126 patients with SLE. All of them were tested for serum IgA, tTG-IgA, DGPs, and EMA. None of the patients were previously diagnosed with CD. As a result, they showed that for tTG-IgA, but not for EMA, the prevalence in SLE was higher than in the general population. No correlation of DGP with any lupus clinical features was identified [111].

## 6. Conclusions

GI involvement is rare and little described in cSLE. Every part of the intestinal tube and every organ of the digestive system can be affected by cSLE, potentially leading to severe complications that have an impact on morbidity and mortality.

The causes of gastrointestinal involvement are very variable. Diagnosis may be a challenge as different etiologies and pathogenetic mechanisms can be involved.

Abdominal pain is not an uncommon feature in cSLE and the underlying causes are different, ranging from acute pancreatic inflammation to intestinal peritonitis and obstruction. Vasculitis is the most common cause of mucosal damage. A rare complication of the mesenteric small- and medium-sized vessel involvement also includes an impaired intestinal permeability manifesting as hypoalbuminemia, edema, ascites, and diarrhea.

Finally, different drugs could be responsible for liver damage in cSLE such as methotrexate, hydroxychloroquine, and azathioprine.

For these reasons, in order to provide early diagnosis and proper treatments, multidisciplinary cooperation is often required to better identify and manage this condition.

## Figures and Tables

**Figure 1 children-10-00309-f001:**
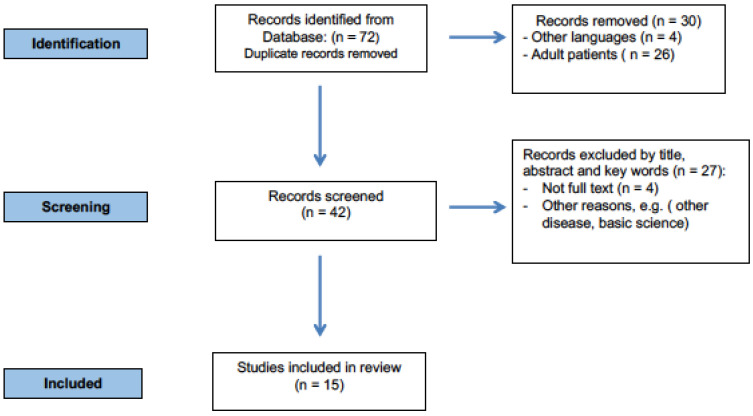
Flowchart of the literature search and study selection process.

**Figure 2 children-10-00309-f002:**
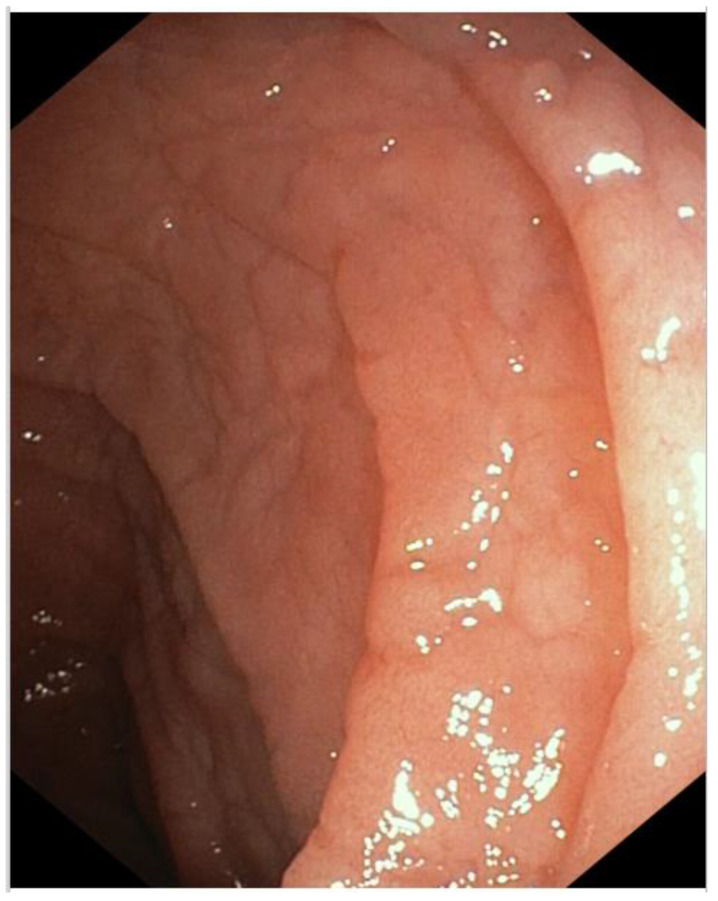
Gastroscopy in patients with cSLE and CD overlap.

**Table 1 children-10-00309-t001:** SLICC CRITERIA.

Clinical Criteria	Immunologic Criteria
Acute cutaneous lupus (macupapular lupus rash, malaria rash, photosensitive lupus rash, etc.)	High ANA concentration
Chronic cuteness lupus (discoid rash, mucosal lupus, etc.)	High anti-dsDNA antibody concentration
Oral or nasal ulcers	Presence of anti-Sm
Nonscarring alopecia	Positive APA
Synovitis involving two or more joints	Low complement (C3, C4, CH50)
Serositis	Direct Coombs test
Renal	
Neurologic (seizure, psychosis, others)	
Hemolytic anemia	
Leukopenia or lymphopenia (without an identifiable cause)	
Thrombocytopenia (without an identifiable cause)	

**Table 2 children-10-00309-t002:** IAIHG Criteria.

Clinical Features	Score
Female gender	+2
ALP: AST ratio - <1.5 - 1.5–3.0 - >3.0	+2 0 −2
Serum globulin or IgG above normal - >2.0 - 1.5–2.0 - 1.0–1.5 - <1.0	+3 +2 +1 0
ANA, SMA, LKM1 - <1:80 - 1:80 - 1:40 - <1:40	+3 +2 +1 0
Illicit drug use history - Positive - Negative	−4 +1
Average alcohol intake daily - <25 g/day - >60 g/day	+2 −2
Histologic findings - Interface hepatitis - Lymphoplasmacytic infiltrate - Rosette formation - None of the above - Biliary changes - Other changes	+3 +1 +1 −5 −3 +2
Other autoimmune disease	+2
AMA positivity	−4
Hepatitis viral markers - Positive - Negative	−3 +3
Aggregate score without treatment - Definite AIH - Probable AIH	>151 0–15

## Data Availability

Not applicable.

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
