# Peer review of "Gastrointestinal Involvement in Children with Systemic Lupus Erythematosus"

_children, 2023, doi:10.3390/children10020309_

Round 1

Reviewer 1 Report

The introduction is well written and no corrections are required. The article was well written, good language to read, great review, I just though they could have more recent articles, for example, follow PMID: 30641536. 

Author Response

Reviewer 1: The introduction is well written and no corrections are required. The article was well written, good language to read, great review, I just though they could have more recent articles, for example, follow PMID: 30641536.

We added more recent articles including PMID: 30641536.

Reviewer 2 Report

The introduction is too long and confusing. Studies with a lot of numerical data are listed. 

On page 2, in the text on liver damage, the results of research on liver damage in adults are presented, which cannot be compared with changes in children.

Manifestations of GI changes in adults are described again on page 4.

I would emphasize that the title of the article refers to children.

On page 3, the description of the patient is given too much space for an article that should be a review.

Author Response

Reviewer 2:

The introduction is too long and confusing. Studies with a lot of numerical data are listed. 

We re-writed the introduction and deleted part of numerical data

On page 2, in the text on liver damage, the results of research on liver damage in adults are presented, which cannot be compared with changes in children. Manifestations of GI changes in adults are described again on page 4.

We deleted the results of some studies in adults.

I would emphasize that the title of the article refers to children. 

We changed the title of the article in “Gastrointestinal Involvement in children with Systemic Lupus Erythematosus”

On page 3, the description of the patient is given too much space for an article that should be a review.

We re-writed and deleted the description of the patient

Reviewer 3 Report

Dear Authors

Please see the commented file as an attachment to make some revisions. Also, the manuscript lacks a good conclusion.

Sincerely,

Author Response

Comment 1: Line 33 we changed in "treatment adverse effect"

Comment 2: we checked and correct a duplicate sentences

Commnt 3: we corrected the sentence 

Comment 4: We added citation

Comment 5: we added citation

Comment 6: We revised the sentence

Comment 7: We added a “Conclusion” paragraph